# Multi-Target Localization Based on Unidentified Multiple RSS/AOA Measurements in Wireless Sensor Networks

**DOI:** 10.3390/s21134455

**Published:** 2021-06-29

**Authors:** Seyoung Kang, Taehyun Kim, Wonzoo Chung

**Affiliations:** 1Department of Computer Science and Engineering, Korea University, Seoul 02841, Korea; sykang0229@korea.ac.kr; 2Agency for Defense Development, Daejeon 34186, Korea; thkimc@hanmail.net; 3Department of Artificial Intelligence, Korea University, Seoul 02841, Korea

**Keywords:** wireless sensor networks (WSNs), multi-target, localization, received signal strength (RSS), angle of arrival (AOA)

## Abstract

All existing hybrid target localization algorithms using received signal strength (RSS) and angle of arrival (AOA) measurements in wireless sensor networks, to the best of our knowledge, assume a single target such that even in the presence of multiple targets, the target localization problem is translated to multiple single-target localization problems by assuming that multiple measurements in a node are identified with their originated targets. Herein, we first consider the problem of multi-target localization when each anchor node contains multiple RSS and AOA measurement sets of unidentified origin. We propose a computationally efficient method to cluster RSS/AOA measurement sets that originate from the same target and apply the existing single-target linear hybrid localization algorithm to estimate multiple target positions. The complexity analysis of the proposed algorithm is presented, and its performance under various noise environments is analyzed via simulations.

## 1. Introduction

Wireless sensor networks (WSNs) are spatially spread sensor nodes for monitoring or detecting physical environments [1,2,3,4,5,6,7,8]. Recently, outstanding developments in linear estimation algorithms have been realized for the localization of a target using both received signal strength (RSS) and angle of arrival (AOA) measurements of sensors in WSNs, i.e., a feature that was not originally designed for target localization [9,10]. Since target localization becomes extremely difficult when line-of-sight assumption cannot be applied, for example, severe multi-path environment [11,12], the existing studies, including this paper, assume a line-of-sight environment.

Most of these localization algorithms assume that anchor nodes contain a unique measurement set from a single target. However, in the presence of multiple targets, multiple measurement sets at each anchor node exist, which complicates the estimation process of the targets. Most existing studies regarding multi-target detection in WSNs assume that the targets are sparsely located over networks and, therefore, each anchor node contains a single set of RSS/AOA measurements from a single target. Various multi-target localization methods that exploit the network topology [13,14,15,16,17,18,19,20,21,22,23,24] or apply compressive sensing [25,26,27] have been proposed. Some existing studies assume that each anchor node can identify the origin of the RSS/AOA measurements using, for example, code division multiplexing access codes [28,29,30,31,32,33,34,35,36], therefore solving multiple single-target localization problems.

However, when RSS/AOA measurements are not identifiable by their originated targets, for example, in non-cooperative situations such as military applications, the existing linear estimation schemes are not directly applicable. A straightforward grid-search approach using the maximum-likelihood function can be considered. However, the computational complexity of this approach is prohibitive to increased numbers of anchor nodes and targets, particularly for an increased number of grids.

Herein, we propose a computationally efficient scheme for multi-target localization based on a linear estimation method. First, we identify all possible target candidates by applying a single-target linear estimation algorithm for all possible combinations of the measurement sets in minimally selected initial anchor nodes (e.g., 1∼3) and identify tentative target positions by evaluating the simplified maximum-likelihood cost function. Subsequently, for a tentative target, we determine the RSS/AOA measurement set based on the minimum approximated likelihood cost in each node. Finally, the final target positions are estimated based on the clustered RSS/AOA measurements using all available anchor nodes via a single-target linear estimation method. The proposed approach exhibits polynomial complexity $O(M^{N}N)$ with respect to the number of targets, *M*, and the number of anchor nodes, *N*. Simulation results confirm the robust performance of the proposed algorithm, which inherits the performance of the single-target algorithm in the presence of severe noise. In summary, the main contribution of this work is to present a computationally efficient localization scheme for multiple targets based on RSS and AOA measurements that are not identified with their originated targets. The proposed algorithm presents a novel clustering scheme to identify the measurement sets in each anchor node with the originated target in a polynomial complexity in terms of the number of targets so that the resulting multi-target localization algorithm can be handled by the current computational power regardless of the number of targets.

The remainder of this paper is structured as follows: Section 2 introduces the system model of a multi-target localization problem in WSNs. Section 3 presents the proposed algorithm. In Section 4, the complexity of the proposed algorithm is analyzed. Section 5 presents the performance evaluation of the proposed method, which was conducted via simulation. Finally, Section 6 presents the conclusions of the study.

## 2. System Model and Problem Setting

Figure 1 illustrates a WSN with multiple targets. Let $a_{i}={\left[a_{ix},a_{iy},a_{iz}\right]}^{T}\in R^{3}$ be the known position of the *i*-th sensor (anchor) node for i=1,⋯,N and $x_{m}={\left[x_{mx},x_{my},x_{mz}\right]}^{T}$ be the unknown position of the *m*-th sensor (target) for m=1,⋯,M. We assume that all anchor nodes have array antenna or directional antenna to measure AOA (azimuth and elevation angle) [37] via the line-of-sight (LoS) path from the target [13,14,15,16,17,18,19,20,21,22,23,24]. In the presence of multiple targets, we assume that each anchor node receives signals from the targets, and the RSS, azimuth angle, and elevation angle are measured from these signals. Let us denote the RSS, azimuth angle, and elevation angle from the *m*-th received signal as ${\hat{P}}_{im}$, ${\hat{\phi }}_{im}$, and ${\hat{\alpha }}_{im}$, respectively. It is noteworthy that in general, the *m*-th received signal is not necessarily associated with the *m*-th target.

The RSS can be viewed as a function of the target and anchor node positions using a path-loss model. In the absence of noise, the RSS $\left(P_{im}\right)$ at a distance $d_{im}=||x_{m}-a_{i}||$ is modeled as [38]
(1)$$\begin{array}{c} P_{im}=P_{T}{\left(\frac{d_{0}}{d_{im}}\right)}^{\gamma }10^{-\frac{L_{0}}{10}}, \end{array}$$
where $P_{T}$ is the transmit power of the target, which is assumed to be the same for all targets; and γ denotes the path-loss exponent; $L_{0}$ is the power loss at a reference distance $d_{0}$. Defining $P_{0}:=P_{T}-L_{0}$ and setting $d_{0}=1$ as the unit length, the following noisy model in the decibel (dB) scale is obtained:(2)$$\begin{array}{c} {\hat{P}}_{im}=P_{0}-10\gamma \log_{10}d_{im}+w_{im}, \end{array}$$
where $w_{im}$ is a zero-mean Gaussian noise with variance $\sigma_{w_{im}}^{2}$, i.e., $w_{im}∼N\left(0,\sigma_{w_{im}}^{2}\right)$.

The azimuth and elevation angle are expressed as functions of the target and anchor node positions. In the presence of noise, we have
(3)$$\begin{array}{c} \begin{array}{cc} {\hat{\phi }}_{im} & =\tan^{-1}\left(\frac{x_{my}-a_{iy}}{x_{mx}-a_{ix}}\right)+u_{im}\\ {\hat{\alpha }}_{im} & =\cos^{-1}\left(\frac{x_{mz}-a_{iz}}{||x_{m}-a_{i}||}\right)+v_{im} \end{array}, \end{array}$$
for i=1,⋯,N and m=1,⋯,M, where $u_{im}$ and $v_{im}$ are zero-mean Gaussian noise with variance, i.e., $u_{im}∼N\left(0,\sigma_{u_{im}}^{2}\right)$ and $v_{im}∼N\left(0,\sigma_{v_{im}}^{2}\right)$, respectively.

We assume that each anchor node contains exactly *M* measurements sets {$({\hat{P}}_{im},{\hat{\phi }}_{im},$${\hat{\alpha }}_{im})$}. When every RSS/AOA measurement set $\left\{({\hat{P}}_{im},{\hat{\phi }}_{im},{\hat{\alpha }}_{im})\right\}$ is identifiable by the originated target $x_{m}$, the multi-target localization problem can be solved based on *M* single-target localization problems. Several intensive studies regarding linear single-target localization [13,14,15,16,17,18,19,20,22,23] exist, and the most well-known approach is to consider the following linear relationship between the target position and RSS/AOA measurements:(4)$$\begin{array}{c} \begin{array}{cc} \lambda_{im}u_{im}^{T}\left(x_{m}-a_{i}\right)-\eta d_{0} & =\varepsilon_{im1}\\ c_{im}^{T}\left(x_{m}-a_{i}\right) & =\varepsilon_{im2}\\ {\left({𝟙}^{T}-u_{im}\cos\left({\hat{\alpha }}_{im}\right)\right)}^{T}\left(x_{m}-a_{i}\right) & =\varepsilon_{im3} \end{array},\;\mathrm{for}\;i=1\cdots N \end{array}$$
where
(5)$$\begin{array}{ccc} \lambda_{im} & := & 10^{\frac{{\hat{P}}_{im}}{10\gamma }},\eta :=10^{\frac{P_{0}}{10\gamma }},1:={\left[0,0,1\right]}^{T}, \end{array}$$
(6)$$\begin{array}{ccc} c_{im} & := & {\left[-\sin\left({\hat{\phi }}_{im}\right),\cos\left({\hat{\alpha }}_{im}\right),\;0\right]}^{T}, \end{array}$$
(7)$$\begin{array}{ccc} u_{im} & := & {\left[\cos\left({\hat{\phi }}_{im}\right)\sin\left({\hat{\alpha }}_{im}\right),\sin\left({\hat{\phi }}_{im}\right)\sin\left({\hat{\alpha }}_{im}\right),\cos\left({\hat{\alpha }}_{im}\right)\right]}^{T}, \end{array}$$
and $\varepsilon_{imj}$ denotes parameter errors.

In a matrix form, we have
(8)$$\begin{array}{c} A_{m}x_{m}-b_{m}=\varepsilon_{m}, \end{array}$$
where
(9)$$\begin{array}{c} A_{m}=\left[\begin{array}{c} \lambda_{1m}u_{1m}^{T}\\ \vdots \\ \lambda_{Nm}u_{Nm}^{T}\\ c_{1m}^{T}\\ \vdots \\ c_{Nm}^{T}\\ {\left(\cos\left({\hat{\alpha }}_{1m}\right)u_{1m}-1\right)}^{T}\\ \vdots \\ {\left(\cos\left({\hat{\alpha }}_{Nm}\right)u_{Nm}-1\right)}^{T} \end{array}\right],\;b_{m}=\left[\begin{array}{c} {\hat{\lambda }}_{1m}u_{1m}^{T}a_{1}+\eta d_{0}\\ \vdots \\ {\hat{\lambda }}_{Nm}u_{Nm}^{T}a_{N}+\eta d_{0}\\ c_{1m}^{T}a_{1}\\ \vdots \\ c_{Nm}^{T}a_{N}\\ {\left(\cos\left({\hat{\alpha }}_{1m}\right)u_{1m}-1\right)}^{T}a_{1}\\ \vdots \\ {\left(\cos\left({\hat{\alpha }}_{Nm}\right)u_{Nm}-1\right)}^{T}a_{N} \end{array}\right], \end{array}$$
and $\varepsilon_{m}={\left[\begin{array}{ccccccccc} \varepsilon_{1m1} & \cdots & \varepsilon_{Nm1} & \varepsilon_{1m2} & \cdots & \varepsilon_{Nm2} & \varepsilon_{1m3} & \cdots & \varepsilon_{Nm3} \end{array}\right]}^{T}$. Three unknowns ($x_{mx}$, $x_{my}$, $x_{mz}$) exist in (Equation 4), and three equations exist for the single anchor node. Hence, theoretically, only a single anchor node is sufficient to estimate the target position. Therefore, Equation (Equation 8) is solvable for $A_{m}$ and $b_{m}$ constructed using measurement sets $\left\{({\hat{P}}_{im},{\hat{\phi }}_{im},{\hat{\alpha }}_{im})\right\}$ for an arbitrary number of anchor nodes. Because the noise mitigation for the target estimation depends on the number of anchor nodes, the more anchor nodes used, the more accurate is the estimated target position.

The generalized least-squares solution of Equation (Equation 8) is expressed as [39]
(10)$$\begin{array}{c} {\hat{x}}_{m}={\left(A_{m}^{T}W_{m}^{-1}A_{m}\right)}^{-1}A_{m}^{T}W^{-1}b_{m}, \end{array}$$
where $W_{m}=\mathrm{Cov}\left(\varepsilon_{m}|A_{m}\right)$. Several weighted least square (WLS) solutions for approximating $W_{m}$ exist, e.g., the target-range WLS solution uses range-based weights in [19], and error covariance WLS (EC-WLS) uses the approximated error covariance matrix shown in [22]. The EC-WLS achieves state-of-the-art target estimation accuracy performance in terms of mean squared error (MSE) [22].

However, when *M* RSS/AOA measurement sets in each anchor node are not identifiable by their originated targets, then *M* linear equations corresponding to *M* targets cannot be established separately based on the measurements. In fact, we have $M^{N}$ combinations of measurement sets by selecting one measurement set from each anchor node. Subsequently, $M^{N}$ target positions can be estimated using a linear single-target localization algorithm applied to each possible combination, among which only *M* target positions are true; this results in computational complexity of $O(M^{N}N)$.

Figure 2 illustrates the possible existence of ghost targets for N=2 and M=2 in a two-dimensional space. In this example, four target candidates are involved, among which two are real targets and two are ghost targets. To remove the ghost targets and identify the real targets, a certain cost function for each *M* tuples of candidate targets among the candidate targets must be evaluated; this requires additional complexity $O(M^{N}H_{M})$, where *H* denotes the number of combinations with replacement.

Herein, we propose a computationally efficient multi-target localization method that exhibits total polynomial complexity of *M* and *N*.

## 3. Proposed Method

Multi-target localization with unidentified measurement sets is subject to an exponential complexity, $O(M^{N})$. To reduce the complexity, we first estimate $M^{K}$ target position candidates based on the measurements from arbitrary initial *K* anchor nodes (K=1,2,or3≪N) using an existing single-target localization algorithm $M^{K}$ times for all possible combinations of the measurements sets. Subsequently, we identify the most probable *M* target candidates that fit the measurement sets among the $M^{K}$ target candidates. For each anchor node, we assign each of the *M* measurements sets to the most likely originating target candidate among the *M* tentative targets. Finally, based on the identified measurement sets in all anchor nodes, we estimate the refined target positions by applying the conventional single-target algorithm *M* times. The detailed processes of each step are described below.

### 3.1. Estimation of Initial Targets

Among *N* anchor nodes, we randomly select initial *K* anchor nodes, where *K* is selected to be substantially less than *N*, K≪N (in practice, 1≤K≤3). Let $M_{K}$ be the set of all combinations of *K*-tuple measurements constructed by selecting one set of measurements from the distinct *K* anchor nodes. A total of $M^{K}$ tuples exist, and let *l* denote the index for the tuple, $l=1,\cdots ,M^{K}$, and
(11)$$\begin{array}{c} M_{K}:=\left\{\left[\left({\hat{P}}_{l_{1}},{\hat{\phi }}_{l_{1}},{\hat{\alpha }}_{l_{1}}\right),\cdots ,\left({\hat{P}}_{l_{K}},{\hat{\phi }}_{l_{K}},{\hat{\alpha }}_{l_{K}}\right)\right]|l=1,\cdots ,M^{K}\right\}, \end{array}$$
where the index $l_{k}\in \left\{1,\cdots ,M\right\}$ denotes the measurement set index of the *l*-th measurement tuple in the *k*-th anchor node. For the *l*-th measurement tuple $[\left({\hat{P}}_{l_{1}},{\hat{\phi }}_{l_{1}},{\hat{\alpha }}_{l_{1}}\right),\cdots ,({\hat{P}}_{l_{K}},$${\hat{\phi }}_{l_{K}},\;{\hat{\alpha }}_{l_{K}})]$, a single-target estimation method such as EC-WLS algorithm [22] is applied, as if they originated from the same target, and let $x_{l}$ denote the estimated target candidate position corresponding to the *l*-th tuple. Among $M^{K}$ target candidates, only *M* targets are true, and the remaining target candidates are ghost targets. To determine the true *M* targets that fit the measurement data, we considered all possible *M* combinations of distinct target candidates. Let $X_{MK}$ denote the set of all *M*-tuples of distinct target estimates from $\left\{x_{l}\right\}$, i.e.,
(12)$$\begin{array}{c} X_{MK}=\{{\vec{x}}_{q}:=\left[x_{q_{1}},\cdots ,x_{q_{M}}\right]|q=1,\cdots {,}_{M^{K}}C_{M},q_{m}\in \left\{1,\cdots ,M^{K}\right\}\mathrm{and}\;q_{i}<q_{j}\;\mathrm{if}\;i<j\} \end{array}$$

It is noteworthy that the size of $X_{MK}$ is $M^{K}C_{M}$.

Ideally, it is desirable to determine the best *M* tentative target positions to fit the data based on maximum log-likelihood principle.
(13)$$\begin{array}{c} \left[x_{1}^{*},\cdots ,x_{M}^{*}\right]=\arg\max_{{\vec{x}}_{q}\in X_{MK}}\ln P\left(M_{K}|{\vec{x}}_{q}\right), \end{array}$$
where $\ln P(M_{K}|{\vec{x}}_{q})$ is the log-likelihood function of ${\vec{x}}_{q}$ for the measurement data $M_{K}$. Because the exact $\ln P(M_{K}|{\vec{x}}_{q})$ is difficult to compute, we propose an approximating cost function based on the following approximations:(i)Assuming that the measurement errors are independently and identically distributed Gaussian (which we have already assumed), the log-likelihood function $\ln P(M_{K}|{\vec{x}}_{q})$ for the measurement set $M_{K}$ for a target vector ${\vec{x}}_{q}$ can be written as
(14)$$\begin{array}{c} \ln P\left(M_{K}|{\vec{x}}_{q}\right)=-\sum_{l=1}^{M^{K}}\sum_{k=1}^{K}\sum_{m=1}^{M}\left(\frac{1}{\sigma_{w_{l_{k}}}^{2}}||{{\hat{P}}_{l_{k}}-P_{kx_{q_{m}}}||}^{2}+\frac{1}{\sigma_{u_{l_{k}}}^{2}}||{{\hat{\phi }}_{l_{k}}-\phi_{kx_{q_{m}}}||}^{2}+\frac{1}{\sigma_{v_{l_{k}}}^{2}}||{{\hat{\alpha }}_{l_{k}}-\alpha_{kx_{q_{m}}}||}^{2}\right)+\;C, \end{array}$$
where $P_{kx_{q_{m}}}$, $\phi_{kx_{q_{m}}}$, and $\alpha_{kx_{q_{m}}}$ are the computed RSS and AOA from the position ($x_{q_{m}}$) and the anchor node ($a_{k}$), respectively, and *C* is a constant from the multivariate Gaussian pdf.(ii)Furthermore, assuming $\sigma_{w_{lk}}=\sigma_{u_{lk}}=\sigma_{v_{lk}}$, we define a cost function based on Euclidean distance between the measurement data $M_{K}$ and the synthesized measurements from the target candidate ${\vec{x}}_{q}$ and *K* anchors.
(15)$$\begin{array}{c} J\left({\vec{x}}_{q};M_{K}\right)=\sum_{l=1}^{M^{K}}\sum_{k=1}^{K}\sum_{m=1}^{M}\left(||{{\hat{P}}_{l_{k}}-P_{kx_{q_{m}}}||}^{2}+||{{\hat{\phi }}_{l_{k}}-\phi_{kx_{q_{m}}}||}^{2}+||{{\hat{\alpha }}_{l_{k}}-\alpha_{kx_{q_{m}}}||}^{2}\right) \end{array}$$

The target estimates ${\vec{x}}^{*}$ minimizing this quadratic cost function is the best multi-target positions; it simultaneously fits the measurement data $M_{K}$ in terms of mean squared error and maximizes maximum likelihood. The cost function requires $M^{K}C_{M}M^{K}KM$ evaluations of the distance between the measured and synthesized data and can be slightly modified by exploiting the minimum searching structure that yields $M_{K}C_{M}M!KM$ evaluations.
(16)$$\begin{array}{c} \begin{array}{cc} {\vec{x}}^{*} & =\arg\min_{{\vec{x}}_{q}\in X_{MK}}J\left({\vec{x}}_{q};M_{K}\right)\\ & =\arg\min_{{\vec{x}}_{q}\in X_{MK}}\sum_{k=1}^{K}\min_{\vec{a}\in P\left(M\right)}\sum_{m=1}^{M}\left(||{{\hat{P}}_{ka_{p}}-P_{kx_{q_{m}}}||}^{2}+||{{\hat{\phi }}_{ka_{p}}-\phi_{kx_{q_{m}}}||}^{2}+||{{\hat{\alpha }}_{ka_{p}}-\alpha_{kx_{q_{m}}}||}^{2}\right) \end{array} \end{array}$$
where P(M) is the set of all permutations of (1,⋯,M), i.e.,
(17)$$\begin{array}{c} P\left(M\right)=\left\{[p_{1},\cdots ,p_{M}\right]|p_{i}\in \left\{1,\cdots ,M\right\}\;\mathrm{and}\;p_{i}\neq p_{j}\} \end{array}$$

To substantially reduce the computational complexity, we focus on a single-target candidate $x_{l}\in X_{MK}$ individually, instead of an *M* tuple simultaneously. In the absence of noise, for a true target, each node should have exactly one measurement set that matches the synthesized measurement with the true target. Let $d_{l}$ denote the sum of the minimum distance between the measurement data sets in a node and the synthesized measurement for all nodes.
(18)$$\begin{array}{c} d_{l}=\sum_{k=1}^{K}\min_{m\in \left\{1,\cdots ,M\right\}}\left(||{{\hat{P}}_{km}-P_{kx_{l}}||}^{2}+||{{\hat{\phi }}_{km}-\phi_{kx_{l}}||}^{2}+||{{\hat{\alpha }}_{km}-\alpha_{kx_{l}}||}^{2}\right), \end{array}$$

Let $d_{l_{1}},\cdots ,d_{l_{M}}$ be the *M* smallest distance among all $\left\{d_{l}\right\}$. The corresponding target candidates, $x_{l_{1}},\cdots ,x_{l_{M}}$, are the estimated tentative targets in this one-by-one approach. This one-by-one approach may not be optimal, but it significantly reduces the computational complexity to $M^{K+1}K$ evaluations.

### 3.2. Measurements Clustering Based on Initial Targets

Based on the tentative targets $\left[x_{1}^{*},\cdots ,x_{M}^{*}\right]$ obtained via block-by-block approach (Equation 16) or one-by-one approach (Equation 18), we determined the measurement sets associated with each tentative target $x_{m}^{*}$. Additionally, we considered two different approaches: the block-by-block approach and the one-by-one approach.

(i)Block-by-block approach: For a block of ordered *M* tentative targets $\left[x_{1}^{*},\cdots ,x_{M}^{*}\right]$, for each node i=1,⋯,N, we associate ordered *M* indexes $\left[p_{i1}^{*},\cdots ,p_{iM}^{*}\right]$ that indicate the order of measurement set corresponding to the tentative targets in the *i*-th anchor node as following:
(19)$$\begin{array}{c} \left[p_{i1}^{*},\cdots ,p_{iM}^{*}\right]=\arg\min_{\vec{a}\in P\left(M\right)}\sum_{p=1}^{M}\left(||{{\hat{P}}_{ia_{p}}-P_{ix_{p}^{*}}||}^{2}+||{{\hat{\phi }}_{ia_{p}}-\phi_{ix_{p}^{*}}||}^{2}+||{{\hat{\alpha }}_{ia_{p}}-\alpha_{ix_{p}^{*}}||}^{2}\right). \end{array}$$
where P(M) denotes the set of all permutations of 1,⋯,M and $\vec{a}$ denotes an element of P(M).(ii)One-by-one approach: For each target candidate $x_{m}^{*}$, we associate a best fitting measurement set index, denoted by $p_{im}^{*}$, in the *i*-th anchor node for i=1,⋯,N, as following:
(20)$$\begin{array}{c} p_{im}^{*}=\arg\min_{a\in \left\{1,\cdots ,M\right\}}\left(||{{\hat{P}}_{ia}-P_{ix_{m}^{*}}||}^{2}+||{{\hat{\phi }}_{ia}-\phi_{ix_{m}^{*}}||}^{2}+||{{\hat{\alpha }}_{ia}-\alpha_{ix_{m}^{*}}||}^{2}\right), \end{array}$$

Once all measurement sets are clustered with respect to the corresponding tentative targets, we have *M* sets of measurement sets containing *N* measurement sets over anchor node *N*: (21)$$\begin{array}{c} \left\{({\hat{P}}_{ip_{im}^{*}},{\hat{\phi }}_{ip_{im}^{*}},{\hat{\alpha }}_{ip_{im}^{*}})|i=1,\cdots ,N\right\}\;\;\mathrm{for}\;m=1,\cdots M \end{array}$$

### 3.3. Final Target Estimation Based on Measurements Clustering

Using the clustered measurement sets (Equation 21), we finally refine $x_{m}^{*}$ by applying a single-target localization algorithm, e.g., EC-WLS algorithm [22], based on *N* measurement sets instead of only *K* measurement sets. Let ${\hat{x}}_{m}$ denote the final EC-WLS solution based on the measurement sets $\left\{({\hat{P}}_{ip_{im}^{*}},{\hat{\phi }}_{ip_{im}^{*}},{\hat{\alpha }}_{ip_{im}^{*}})|i=1,\cdots ,N\right\}$,
(22)$$\begin{array}{c} {\hat{x}}_{m}={\left(A_{m}^{T}W_{m}^{-1}A_{m}\right)}^{-1}A_{m}^{T}W_{m}^{-1}b_{m},\;m=1,\cdots ,M \end{array}$$
where $A_{m}$, $b_{m}$, and $W_{m}$ are matrices constructed using the measurement sets with $x_{m}^{*}$, $\left\{({\hat{P}}_{ip_{im}^{*}},{\hat{\phi }}_{ip_{im}^{*}},{\hat{\alpha }}_{ip_{im}^{*}})|i=1,\cdots N\right\}$ obtained from [22].

## 4. Complexity Analysis

Herein, we summarize the complexities of the proposed multi-target localization schemes. In the first step, we randomly select *K* anchor nodes and identify the best *M* target candidates based on the measurements from *K* anchor nodes using the approximated ML cost function. The block-by-block approach requires $O(M^{K}P_{M}K)$, whereas the one-by-one approach requires $O(M^{K+1}K)$. In the second step, we cluster the measurements sets after identifying the best *M* target candidates. The block-by-block approach requires O(NM!), whereas the one-by-one approach requires $O(NM^{2})$. Finally, each target is refined by a single-target localization algorithm, such as EC-WLS [22]. The block-by-block and one-by-one approaches have the same complexity, O(NM).

As summarized in Table 1, the complexity of both methods is linear with respect to the node size *N* and increases exponentially with respect to *K* as a factor of *M*. Based on $M^{K}P_{M}=M^{K}!/\left(M^{K}-M\right)!\approx M^{KM}$, it is evident that the complexity of the block-by-block approach increases exponentially as the target size *M* increases, whereas the complexity of the one-by-one approach is primarily affected by $M^{K+1}$ term. Therefore, for large target numbers, the one-by-one approach is recommended.

The *K* selected determines the trade-off between the complexity of the algorithm and the accuracy of multi-target estimation in the presence of noise, as shown in Section 5, via simulation. A large *K* reduces the clustering error rate but increases the complexity, particularly in the block-by-block approach.

## 5. Simulation Results

We have used MATLAB R2020a to perform all simulations using the computer with Intel(R) Core(TM) i5-8600 CPU @ 3.10 GHz. All simulations parameters are set as follows. To generate the measurements, we used Equations (Equation 2) and (Equation 3). The number of Monte Carlo runs ($M_{c}$) is set to 50,000. The reference distance ($d_{0}$) is set to 1 m. The true TP ($P_{0}$) is set to −10 dBm. The true PLE (γ) is set to 2.2 for all anchor nodes. All targets and anchor nodes are randomly located inside a box with an edge length B=10 m. Figure 3 is a deployment example of targets and anchor nodes that are randomly generated under N=6 and M=2.

To evaluate the clustering performance of the proposed algorithms, we define the probability of clustering success (PCS) as follows: (23)$$\begin{array}{cc} P_{cs} & =\frac{\mathrm{total}\;\mathrm{number}\;\mathrm{of}\;\mathrm{targets}\;\mathrm{that}\;\mathrm{correctly}\;\mathrm{clusterred}\;\mathrm{in}\;\mathrm{all}\;\mathrm{nodes}}{\mathrm{total}\;\mathrm{number}\;\mathrm{of}\;\mathrm{targets}}=\frac{1}{MM_{c}}\sum_{i=1}^{M_{c}}\sum_{j=1}^{M}S_{ij},\\ S_{ij} & =\left\{\begin{array}{c} 1,\;p_{kj}^{*}=j\;\mathrm{and}\;p_{kq}^{*}\neq j\;\left(j\neq q\right),\;\mathrm{for}\;k=1,\cdots ,N,\;\mathrm{and}\;q=1,\cdots ,M\\ 0,\;\mathrm{otherwise} \end{array}\right. \end{array}$$
where $S_{ij}$ is indicator function for the successful clustering of the measurement sets over all anchor nodes originating from the *j*-th target at the *i*-th Monte Carlo run defined by analyzing whether the index of the clustered measurements set with respect to the *j*-th target at anchor node *k*, denoted by $p_{kj}^{*}$, is consistent with the true index *j*. The performance of the proposed algorithm shows a trade-off between the accuracy performance measured by PCS or NRMSE and the computational complexity due to the initial node size selected, candidate target selection methods, and clustering methods.

Figure 4 show plots of PCS (a) and normalized root mean square error (NRMSE) (b) based on $\sigma_{w_{im}}=3$ (dB) and $\sigma_{u_{im}}=\sigma_{v_{im}}=5$ (deg) for a fixed number of anchor nodes, N=6, as well as the number of targets M=2 or 3 as the number of initial anchor nodes (*K*) varies from 1 to 3. In Figure 4a, the block-by-block approach demonstrates a better PCS than the one-by-one approach, as expected. However, in terms of NRMSE, both approaches demonstrated similar performance, as shown in Figure 4b.

In fact, the NRMSE is not an ideal indicator for evaluating multi-target localization algorithms for general cases. When targets are located in close proximity to each other, the NRMSE cannot reflect the efficacy of the algorithm in distinguishing the targets because it is bounded by the maximum distance between targets. For moderately sparsely located targets, the NRMSE can be easily disrupted by a single incident of a large error because of false measurement clustering. Therefore, we investigated the performance of the proposed algorithm-based primarily on PCS and analyzed the NRMSE of highly sparsely located targets.

As shown in Figure 5, we investigated the behavior of PCS as the RSS noise increased from 1 to 6 dB for several initial anchor node sizes, K=1,2,3, with fixed angle noises ($\sigma_{u_{im}}=\sigma_{v_{im}}=5$ (deg)) for a fixed target number (M=2) and total anchor node size N=6. As the number of initial anchor node (*K*) increased from 1 to 3 the PCS improved. The PCS depended significantly on the RSS noise when a single initial anchor node (K=1) was used, whereas the deployment of multiple initial anchor nodes (K=2,3) yielded PCS performance that were almost independent of various RSS noise level. This result strongly suggests that a single initial anchor node (K=1) is not suitable for practical implementation, although it simplifies the complexity of the proposed algorithm considerably.

Figure 6 illustrates the resolution performance of the proposed algorithm for a high SNR scenario based on plots of the PCS (a) and NRMSE (b) for various distance between the targets. Two targets (M=2), in which the center was in the origin, were considered, and the distance between the two targets was set from 10 cm to 1 m. The initial target number *K* was set from 2 to 5 for the one-by-one approach and K=2,3 for the block-by-block approach. The noise variances were set to mild RSS $\sigma_{w_{im}}=1$ (dB) and small angle variance $\sigma_{u_{im}}=\sigma_{v_{im}}=0.3$ (deg). At this noise level, the single-target estimation algorithm (EC-WLS) yielded an RMSE of approximately 0.036 when the measurement sets were identified correctly. The PCS by the block-by-block approach outperformed that by one-by-one approach by up to 0.6 m. It is noteworthy that for this high SNR setting, the initial target number K=2,3,4,5 indicated almost similar PCS performance.

Figure 7 illustrates the resolution performance of the proposed algorithm for a low SNR. Two targets (M=2) whose center was located at the origin were considered. The distance between the two targets was set from 10 cm to 5 m. The number of initial anchor node *K* is set from 2 to 5 for the one-by-one approach and K=2,3 for block-by-block approach. The noise variances were set to $\sigma_{w_{im}}=3$ (dB) and $\sigma_{u_{im}}=\sigma_{v_{im}}=5$ (deg). The benign NRMSE performance, which contrasts with the low PCS for a within-target distance of less than 1 m, can be explained by the fact that the MSE due to false classifications is limited to 1 m. This demonstrates that the PCS is a more appropriate indicator for assessing the resolution performance of a multi-target localization algorithm than the NRMSE. It is noteworthy that the PCS and NRMSE performance of K=3 are acceptable compared with those of K=4,5, considering the exponentially increased additional computation burdens.

Figure 8 shows the performance of the proposed algorithm for four targets (M=4) for a low SNR scenario. For M=4, the block-by-block approach becomes impractical because of the exponentially increased complexity; therefore, only the one-by-one approach was considered. The targets were in the vertices of a cube centered at the origin, and the minimum distance among the targets increased from 1 to 10 m. The number of initial anchor nodes was set to K=2,3, and 4 for $\sigma_{w_{im}}=3$ (dB) and $\sigma_{u_{im}}=\sigma_{v_{im}}=5$ (deg). For this low SNR and four-target situation, the PCS deteriorated substantially, but a reasonable NRMSE was obtained for K=3.

Figure 9 shows the NRMSE performance of the proposed algorithm for the number of targets M=2,3 and initial anchor nodes K=2,3 for the one-by-one approach, and initial anchor nodes K=2 for block-by-block approach under various RSS noise ($\sigma_{w_{im}}$) from 1 to 6 dB. The angle noise level was set to $\sigma_{u_{im}}=\sigma_{v_{im}}=5$ (deg) with N=6 anchor nodes. In this setting, for M=2, one-by-one approach with K=3 and block-by-block approach with K=2 demonstrated similar NRMSE performance, which approached the single-target localization performance.

Figure 10 shows a plot of the NRMSE performance by the one-by-one approach for number of initial anchor nodes K=2,3 and number of targets M=2,3 under various RSS from 1 to 6 dB. The angle noise was set to $\sigma_{u_{im}}=\sigma_{v_{im}}=5$ (deg) with number of anchor nodes, N=6. As shown in Figure 10, as the *M* increased, the performance deteriorated slightly; however, it was evident that the one-by-one approach performed reasonably. In general, the one-by-one approach with K=3 demonstrated satisfactory performance with reasonable computational complexity.

## 6. Conclusions

Herein, we presented a computationally efficient multi-target localization scheme that uses a hybrid single-target localization algorithm. A group of target candidates was estimated based on combinations of measurements in a few selected anchor noes, and ghost targets were removed based on the MSE criterion. After obtaining the best *M* target candidates, the proposed algorithm clustered the measurement sets in each anchor node with respect to the target candidates. Finally, a single-target localization algorithm, such as EC-WLS, was applied to the clustered measurement sets in every node to refine the target position. Two different approaches, i.e., block-by-block and one-by-one, were proposed for the initial target estimation phase and measurement clustering phase, respectively. The complexity and resolution performance of the proposed methods with respect to parameter settings were investigated via analysis and simulations. In this paper, we have assumed that no prior information on the targets is given. In the presence of the prior information such as the distance among targets or rough prediction of target positions, the complexity of the proposed algorithm can be reduced by ruling out less probable ghost targets and consequently the overall accuracy performance can be improved, which is the scope for the future research.

## Figures and Tables

**Figure 1 sensors-21-04455-f001:**
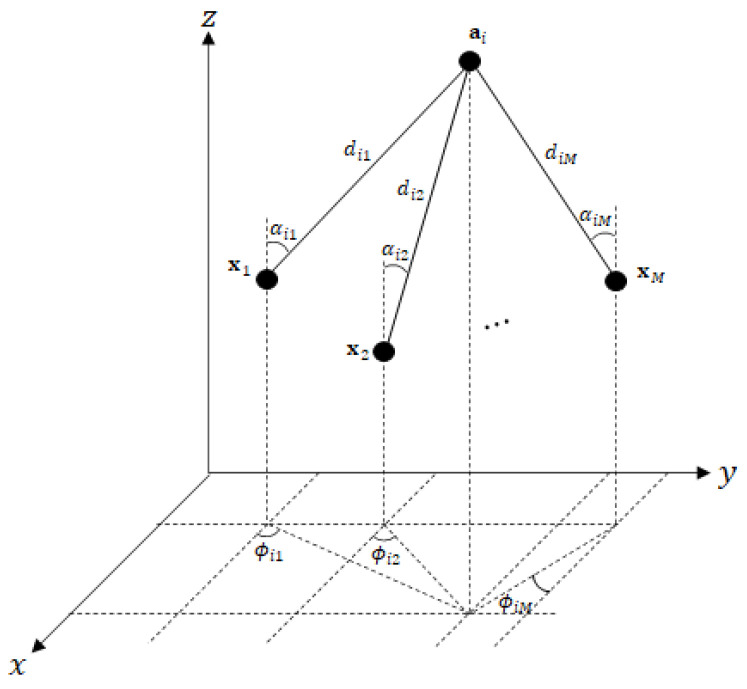
The relationship among an anchor and multi-target in a 3-D space.

**Figure 2 sensors-21-04455-f002:**
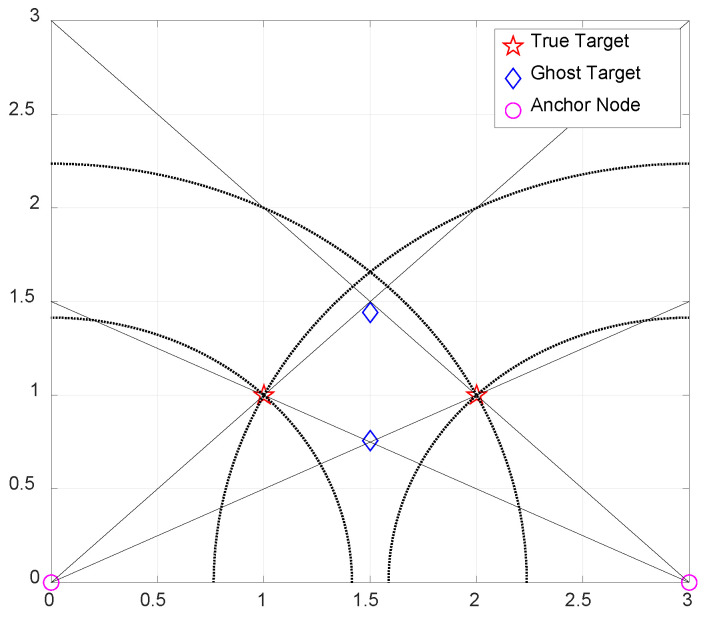
The relationship among true targets and ghost targets in a 2-D space.

**Figure 3 sensors-21-04455-f003:**
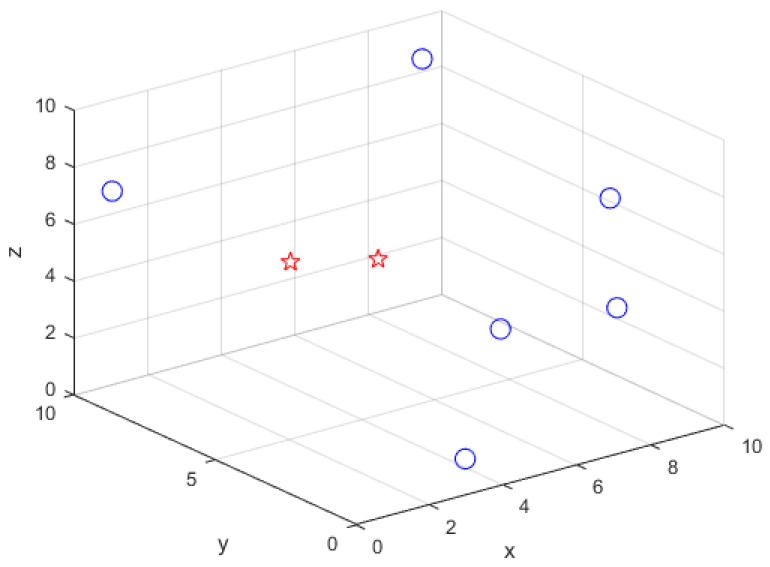
A 3-D deployment example of targets and anchor nodes under N=6 and M=2 where the target and anchor node are represented as the star and circle marks, respectively.

**Figure 4 sensors-21-04455-f004:**
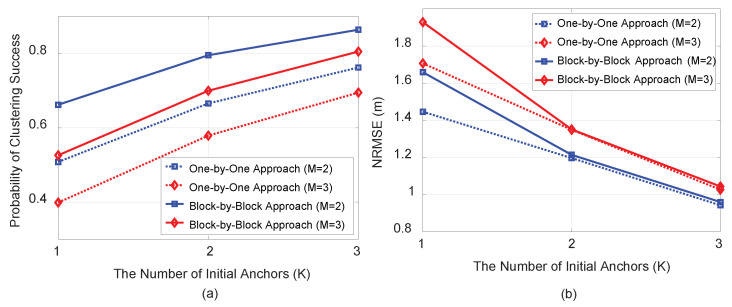
PCS (**a**) and NRMSE (**b**) vs. the number of initial anchor nodes (*K*) under $\sigma_{w_{im}}=3$ (dB) and $\sigma_{u_{im}}=\sigma_{v_{im}}=5$ (deg) with N=6.

**Figure 5 sensors-21-04455-f005:**
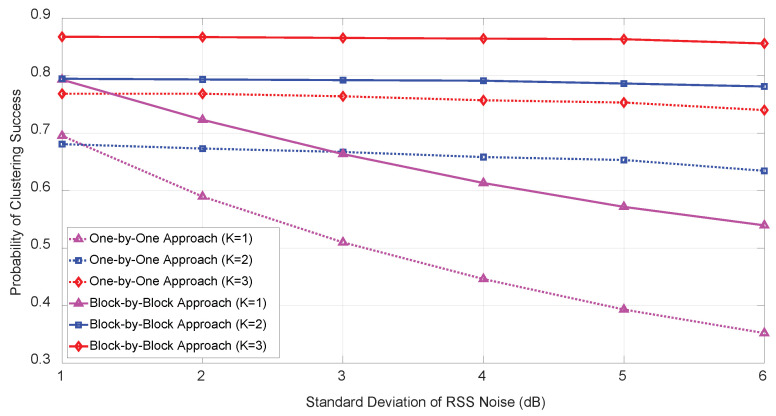
PCS vs. standard deviation of RSS noise ($\sigma_{w_{im}}$) for M=2 and N=6 under $\sigma_{u_{im}}=\sigma_{v_{im}}=5$ (deg).

**Figure 6 sensors-21-04455-f006:**
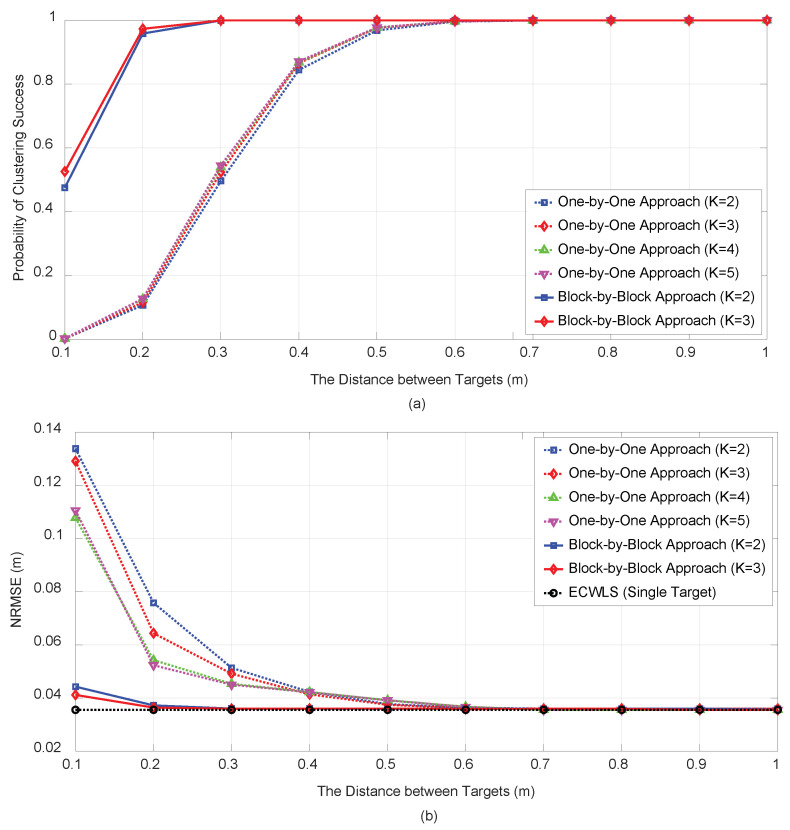
PCS (**a**) and NRMSE (**b**) vs. within-target distance with N=6 under $\sigma_{w_{im}}=1$ (dB) and $\sigma_{u_{im}}=\sigma_{v_{im}}=0.3$ (deg).

**Figure 7 sensors-21-04455-f007:**
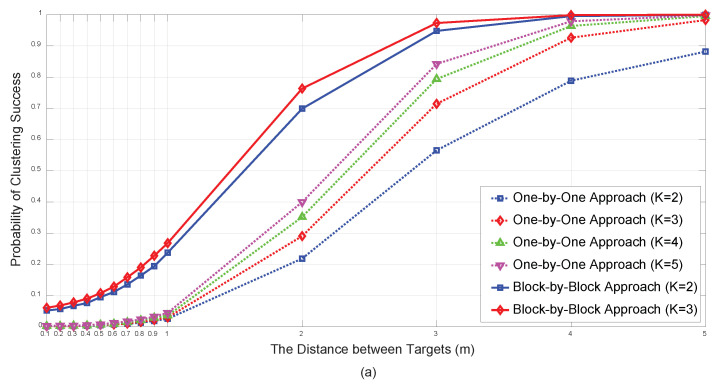
PCS (**a**) NRMSE (**b**) vs. within-target distance for N=6 under $\sigma_{w_{im}}=3$ (dB) and $\sigma_{u_{im}}=\sigma_{v_{im}}=5$ (deg).

**Figure 8 sensors-21-04455-f008:**
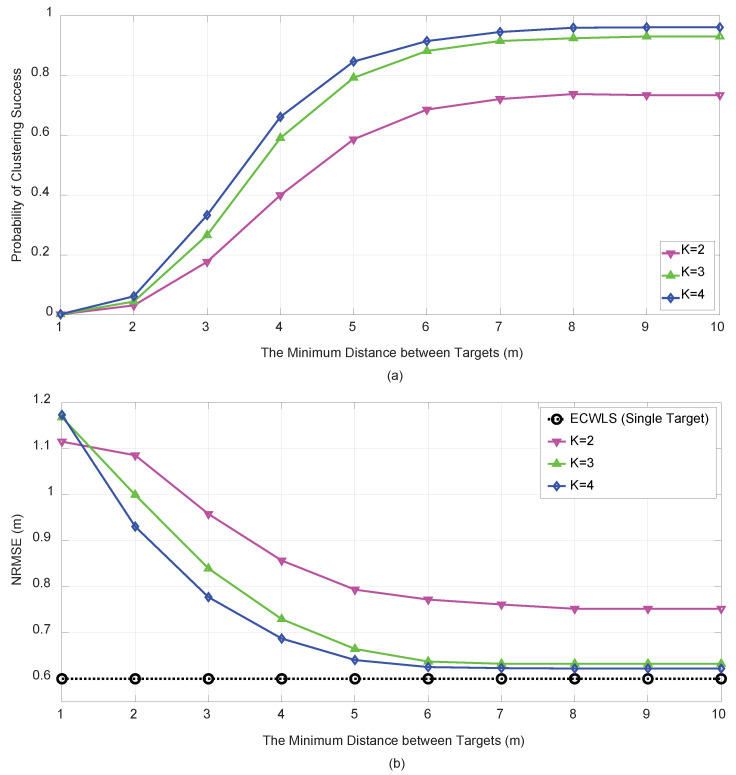
PCS (**a**) and NRMSE (**b**) vs. target minimum distance for M=4 and N=6 under $\sigma_{w_{im}}=3$ (dB) and $\sigma_{u_{im}}=\sigma_{v_{im}}=5$ (deg).

**Figure 9 sensors-21-04455-f009:**
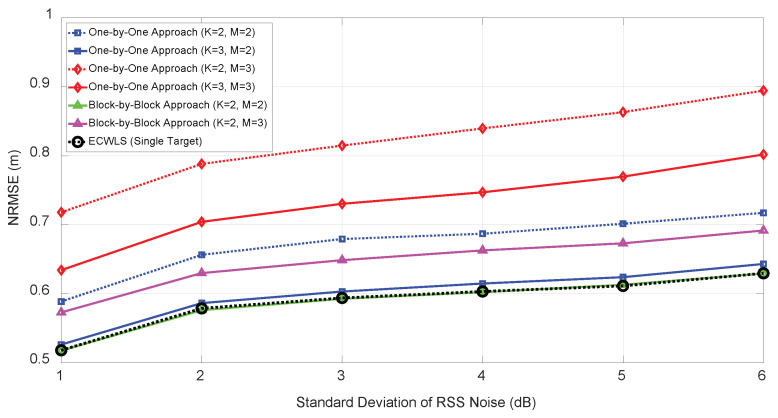
NRMSE according to the number of targets (*M*) and RSS noise ($\sigma_{w_{i}}$).

**Figure 10 sensors-21-04455-f010:**
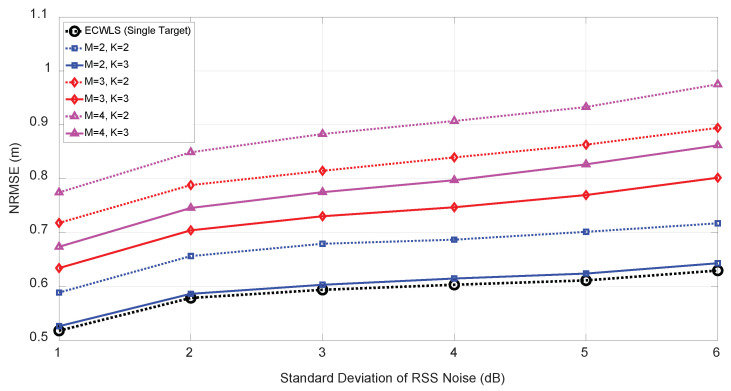
NRMSE vs. the number of targets (*M*) for K=2 and N=6 under $\sigma_{u_{im}}=\sigma_{v_{im}}=5$ (deg).

**Table 1 sensors-21-04455-t001:** Complexity of Proposed algorithm.

Step	One-by-One Approach	Block-by-Block Approach
Estimation of initial targets	$O(M^{K+1}K)$	$O(M^{K}P_{M}MK)$
Measurements clustering	$O(NM^{2})$	O(NM!)
Final target estimation	O(NM)	O(NM)
Total	$O\left(NM+NM^{2}+M^{K+1}K)\right)$	$O\left(NM+NM!+M^{K}P_{M}MK)\right)$

## Data Availability

This study did not report any data.

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
