# Peer review of "Multi-Target Localization Based on Unidentified Multiple RSS/AOA Measurements in Wireless Sensor Networks"

_sensors, 2021, doi:10.3390/s21134455_

Round 1
Reviewer 1 Report
The authors numerically study multi-target localization.
I have multiple major conceptual concerns with the manuscript that should be thoroughly addressed before I can formulate a recommendation:
- The authors fail to specify what radio propagation environment they consider. From the figure it looks like free space. Is this realistic? Please list some representative scenarios in which it is reasonable to assume free space.
- The authors perform simulations without detailing the utilized frequency, the utilized antenna types (unidirectional, directional,...) etc. What signals are emitted by the antennas? All this needs to be explained.
- How can AOA information be available if there is a single measurement antenna as in Fig.1? This is unclear.
- The authors' claim in the first sentence of their abstract appears wrong, I am aware for example of a paper titled "Precise Localization of Multiple Noncooperative Objects in a Disordered Cavity by Wave Front Shaping" in which the measurements are "unidentified". Note also that this paper considers a rich scattering environment, something that the authors completely omit in their review of the localization literature. The same problem of localizing multiple non-cooperative objects has been futher analyzed with artificial intelligence in https://arxiv.org/abs/2103.04711.
- There is a typo in Eq. 23: totla instead of total.
- The authors show plots of the achievable localization precision under various settings, and it is apparent that their is a lower bound to the achievable localization. Can the authors comment on what physical parameters determine this bound? Is it related to the wavelength? I know that in rich scattering environments, there is no fundamental performance limit, and the more reverberation, the better the achievable resolution (https://arxiv.org/abs/2102.05642). But in the authors' case, it is not clear why the lower bound is roughly 0,35m.
- A sketch of the considered setup with the spatial location of transmitting and receiving antennas, as well as the objects, is needed to understand the simulations.
- Can the authors incorporate a priori knowledge into their algorithm? e.g. if it is known that the location of two objects is coupled, or that certain objects move along specific predefined trajectories?
Finally, I believe that a few relevant papers have not been discussed by the authors.
- https://arxiv.org/abs/2006.01779
- https://doi.org/10.1109/MSP.2005.1458287
- https://doi.org/10.1109/JSTSP.2019.2933142
Author Response
The authors would like to thank the anonymous reviewer for your valuable comments and suggestions that greatly improved the quality of this paper.

Reviewer 2 Report
An approach to cluster multiple received signal strength (RSS) and angle of arrival (AOA) measurement sets that originate from the same target and apply the existing single-target linear hybrid localization algorithm to estimate multiple target positions, is proposed in this manuscript.
The manuscript deals with current issues, is well written and contains all the necessary elements. However, the manuscript has several shortcomings that can be corrected.
The Section ‘Introduction’ can be significantly expanded by giving other possibilities and examples of solving the same problem.
In addition, authors should expand the description of their previous research in the subject area
The exact contribution of the proposed algorithm should be cited by the authors. So, what was first proposed by the authors, which has not been known in science so far?
It is necessary to supplement the manuscript with suggestions for improvement the manuscript
Suggestions for improvement the manuscript:
- In the introduction add how others solved the same problem?
- State on which software and computer platform the numerical simulations of the proposed algorithm were performed.
- When an abbreviation is used for the first time in the text, it is necessary to state the full name of the abbreviation (e.g. PCS, NRMSE, (Page 8)...).
- Give more information about the software and practical instructions to the reader so that the proposed algorithm can be tested.
- State the number of Monte Carlo runs used in the experiments.
- State (at the beginning of the manuscript) the exact contribution of the proposed algorithm, i.e. what was first proposed by the authors, which has not been known in science so far?
Author Response

(The authors gave the same response as above.)

Round 2
Reviewer 1 Report
The authors have thoroughly revised their manuscript. I suggest they add a discussion on how to use a priori knowledge in their scheme to the conclusion section.
The setup plot is a bit confusing because it is not clear where the multipath comes from. I see now reflecting objects that may cause multipath.
Author Response
Authors deeply appreciate for the reviewer’s helpful and insightful comments.
